# Enhanced Catalytic Dye Decolorization by Microencapsulation of Laccase from *P. Sanguineus* CS43 in Natural and Synthetic Polymers

**DOI:** 10.3390/polym12061353

**Published:** 2020-06-16

**Authors:** Natalia Lopez-Barbosa, Ana Lucía Campaña, Juan C. Cruz, Nancy Ornelas-Soto, Johann F. Osma

**Affiliations:** 1Department of Electrical and Electronic Engineering, Universidad de los Andes, Cra. 1E No. 19a-40, Bogotá D.C. 111711, Colombia; n.lopez10@uniandes.edu.co (N.L.-B.); al.campana10@uniandes.edu.co (A.L.C.); 2Department of Biomedical Engineering, Universidad de Los Andes, Cra. 1E No. 19a-40, Bogotá D.C. 111711, Colombia; jc.cruz@uniandes.edu.co; 3Laboratorio de Nanotecnología Ambiental, Escuela de Ingeniería y Ciencias, Tecnológico de Monterrey, Monterrey 64849, Mexico; ornel@tec.mx

**Keywords:** azo dyes, decolorization, laccase, microcapsules, *P. Sanguineus* CS43

## Abstract

Polymeric microcapsules with the fungal laccase from *Pycnoporus sanguineus* CS43 may represent an attractive avenue for the removal or degradation of dyes from wastewaters. Microcapsules of alginate/chitosan (9.23 ± 0.12 µm) and poly(styrenesulfonate) (PSS) (9.25 ± 0.35 µm) were synthesized and subsequently tested for catalytic activity in the decolorization of the diazo dye Congo Red. Successful encapsulation into the materials was verified via confocal microscopy of labeled enzyme molecules. Laccase activity was measured as a function of time and the initial reaction rates were recovered for each preparation, showing up to sevenfold increase with respect to free laccase. The ability of substrates to diffuse through the pores of the microcapsules was evaluated with the aid of fluorescent dyes and confocal microscopy. pH and thermal stability were also measured for encapsulates, showing catalytic activity for pH values as low as 4 and temperatures of about 80 °C. Scanning electron microscope (SEM) analyses demonstrated the ability of PSS capsules to avoid accumulation of byproducts and, therefore, superior catalytic performance. This was corroborated by the direct observation of substrates diffusing in and out of the materials. Compared with our PSS preparation, alginate/chitosan microcapsules studied by others degrade 2.6 times more dye, albeit with a 135-fold increase in units of enzyme per mg of dye. Similarly, poly(vinyl) alcohol microcapsules from degrade 1.7 times more dye, despite an eightfold increase in units of enzyme per mg of dye. This could be potentially beneficial from the economic viewpoint as a significantly lower amount of enzyme might be needed for the same decolorization level achieved with similar encapsulated systems.

## 1. Introduction

The discharge of effluents comprising synthetic dyes from the textile industry is a worldwide concern owing to its environmental consequences [1,2]. It is estimated that approximately 5000 tons of synthetic dyes are released every year from the textile industry [3]. Wastewaters from the textile industry are rich in salts and harmful dyes, as evidenced by the significant increase in mutagenic activity of grounds and waters after contact with azo and diazo dyes [4,5]. In addition, textile dyes have a high chemical stability, and consequently tend to remain in the environment for extended periods of time. Typical textile industry effluents exhibit fluctuations in parameters such as chemical oxygen demand, biochemical oxygen demand, pH, color, and salinity [6], which have been thought to be linked to their dye concentration. Thus, removal of dyes from wastewaters before their discharge into the environment is of utmost importance to assure regulatory quality standards for the parameters mentioned above.

Dyes used in the textile industry can be classified in accordance to their chemical structure. In general, any dye has two main components. First, the chromophore, which is responsible for the dye color, and second, the auxochrome, which displays electron transferring capabilities and intensifies the color of the chromophore [7]. Industrial dyes containing azo (–N=N–) chromophores represent 70% of all textile dyes [8]. Over the past three decades, a number of methods have been explored to reduce the dye content in textile wastewaters including physical [9,10], chemical [11], and biological [12,13]. The most effective and environmentally safe methods are those referred as biological [14]. Most of these methods are a combination of aerobic and anaerobic processes; however, anaerobic stages promote the formation of mutagenic and carcinogenic amines after cleavage of azo dyes by azo reductases [15]. Moreover, these reductases exhibit a limited applicability owing to their high specificity towards constrained number of dyes [16].

Laccases (E.C. 1.10.3.2) are multicopper oxidoreductase enzymes able to catalyze the oxidation of multiple substrates through the reduction of oxygen into water [12,17,18]. They can be obtained from both fungi and bacteria, and can oxidize both phenolic and non-phenolic substrates. Fungal laccases are more versatile owing to their high redox potential at the T1 copper and superior affinity toward a wider range of substrates [19]. However, the usage of laccase in the treatment of textile effluents can be expensive owing to the lack of recovery after the treatment.

Immobilization and encapsulation have proven useful to improve thermal, chemical, long-term storage, and conformational stability of free laccases. This approach has also been explored for cyclic operation and improved recyclability [20,21]. One of the preferred methods for immobilization is physical adsorption onto various materials including nanostructured thulium oxide [22], iron oxide nanoparticles [23], silver nanoparticles [24], mesoporous silica [25], and carbon nanotubes [26]. This strategy is, however, limited mainly owing to enzyme losses by desorption from the material in harsh chemical environments. Alternatively, covalent immobilization and encapsulation provide higher enzyme retention, thereby leading to better reusability yields. Encapsulation has mostly been attained through conjugated platforms of silica and hydrogels, as well as biopolymers such as alginate and chitosan [27]. In the case of encapsulation, the main drawback is the mass transfer limitations imposed to substrates and products. This generally lowers the catalytic activity of the preparation on a per mass basis [20].

This work aims at encapsulating laccase from *Pycnoporus sanguineus* CS43 in polymeric microcapsules of alginate/chitosan or poly(styrenesulfonate) (PSS) to evaluate the impact on catalytic activity during the degradation of Congo Red. The diffusion of Congo Red into the microcapsules was monitored by tracking the conversion of 2,2′-azino-bis (3-ethylbenzothiazoline-6-sulfonic acid) diammonium salt (ABTS) via light transmission spectroscopy. Colloidal stability was studied via Zeta potential measurements at both acidic and neutral pH values. Thermal and pH stability of free and encapsulated laccases were measured. Congo Red decolorization was measured spectroscopically. Encapsulated laccases were observed using a scanning electron microscope (SEM) prior to and after the degradation process. Initial reaction rates for each encapsulate were calculated from decolorization time traces. We benchmark our findings against laccases from other fungi. Our findings suggest that microencapsulation is a suitable avenue to more efficient laccase dye decolorization.

## 2. Materials and Methods

### 2.1. Materials and Reagents

All materials were used as received and with no further purification. Sodium alginate, low viscosity chitosan, 2,2′-azino-bis(3-ethylbenzothiazoline-6-sulfonic acid) diammonium salt (ABTS), poly(styrenesulfonate) (PSS), copper(II) sulfate (CuSO4), N-(3-Dimethylaminopropyl)-*N*′-ethylcarbodiimide hydrochloride (EDC), rhodamine B (RhB), and sodium chloride (NaCl) were purchased from Sigma Aldrich (St. Louis, MO, USA). Hydrochloric acid (HCl), sodium hydroxide (NaOH), potassium dihydrogen phosphate (KH_2_PO_4_), potassium hydrogen phosphate (K_2_HPO_4_) N-Hydroxysuccinimide (NHS), and *N,N*-Dimethylformamide (DMF) were purchased from Merck, (Darmstadt, Germany). 91% (w/v) Congo red was purchased from Matheson Coleman & Bell, USA. Hoechst 33342 was acquired from Thermo Fisher Scientific (Waltham, MA, USA).

### 2.2. Laccase Purification

Laccase from *P. Sanguineus* CS43 was obtained from tomato medium, as described elsewhere [20]. In brief, mycelia were removed from the culture supernatant by filtration using two tangential flow filters in series, with pore sizes of 0.5 mm and 0.2 mm, respectively. The obtained laccase cocktail was ultra-filtered using a membrane with a cut-off of 10 kDa.

### 2.3. Enzyme Characterization

Two abundant laccase isoforms (Lac I and Lac II) were purified by ultrafiltration, and characterized via ion exchange (IEX) and hydrophobic interaction chromatography, yielding activities close to 285 U mg^−1^. Molecular weights of Lac I and Lac II were determined by sodium dodecyl sulfate–polyacrylamide gel electrophoresis (SDS-PAGE) and were of 68 kDa and 66 kDa, respectively. Both laccases showed high amino acid sequence similarity (91%), and high thermostability at 50 °C and 60 °C. Half-lives approached 277.7 h and 18 h for Lac I, and 35.8 h and 2.25 h for Lac II. The isoforms oxidized common laccase substrates such as ABTS, 2,6-dimethoxyphenol (DMP), and guaiacol at acidic pH conditions. ABTS was the most efficient substrate, showing high specificity constants of 74.82 mM^−1^ s^−1^ and 36.75 mM^−1^ s^−1^ for Lac I and Lac II, respectively. Michaelis constants (K_m_) at pH 3 were found to be of 6.9 µM and 12.2 µM, respectively [19].

### 2.4. Enzymatic Activity Assay for Laccase

Activity measurements were performed as reported by Niku-Paavola et al. [28]. Phosphate buffer was prepared from KH_2_PO4 and K_2_HPO_4_, and adjusted to the required pH with 1M NaOH or 1M HCl solutions. Spectrophotometric measurements were conducted in a Genesis 10S spectrophotometer (Thermo Fisher Scientific, Waltham, MA, USA) for 2 min at 436 nm. One activity unit was defined as the amount of laccase needed to oxidize 1 µmol of ABTS per minute. Laccase activity was expressed in terms of units per liter (U L^−1^).

### 2.5. Microcapsules Formation

Two types of microcapsules were produced to immobilize the enzyme. Alginate/chitosan microcapsules (AC-C) were prepared by the self-assembly method. In this case, 1 mL of laccase (40,000 U L^−1^) was dissolved in 0.88% (w/v) alginate solution in pH 4.0 buffer and stirred for 3 min. Next, 600 µL of 50 mM CuSO_4_ was added and stirred for 30 min to promote capsules assembly. Finally, 2 mL of 0.053% (w/v) chitosan solution was added and stirred for 30 min.

Poly(styrenesulfonate) microcapsules (PSS-C) were prepared by dissolving 1 mL of laccase (40,000 U L^−1^) in 1.78% (v/v) PSS solution in pH 4.0 buffer. Next, 500 µL of 1.5mM NaCl was added to the solution and subsequently stirred for 30 min.

A control experiment where the microcapsules were synthesized in the absence of laccase (i.e., by replacing laccase with milli Q water) was included in the experimental set to evaluate the impact of the material alone on dye removal.

The obtained solutions were vacuum filtered over 0.22 µm pore nylon filter membranes, and subsequently washed three times with milli Q water. The final amounts of microcapsules were 2.131 g and 0.142 g for AC-C and PSS-C, respectively.

### 2.6. Microcapsules Characterization and Activity Measurements

Effective capsule formation and approximate size distribution were determined by direct observation using a BioBlue binocular microscope (Euromex, Arnhem, The Netherlands) equipped with a 400× magnification lens, and an achromatic objective with 0.65 numerical aperture.

Laccase activity measurements were performed using a Genesis 10S Spectrophotometer (Thermo Fisher Scientific, USA) following the same procedure for free laccase. Experiments for free laccase and encapsulates were conducted at 2000 U L^−1^ of enzyme.

Encapsulated enzyme in AC-C and PSS-C was determined as the difference between free laccase activity and residual enzyme activity on the supernatant after filtration. Measurements were carried in triplicate.

Labelled experiments were carried out to determine the distribution of the enzyme among the capsules. Laccase was labeled with RhB for confocal microscopy visualization. Fluorophore conjugation was achieved in a 40% (v/v) DMF medium, with EDC and NHS as coupling agents to form amide bonds between the carboxyl groups of RhB and the free amines of laccase molecules. Filtration via ultracentrifuge was used for labeled enzyme purification with a cellulose membrane of molecular weight cut-off (MWCO) 10 KDa (Merck, Darmstadt, Germany). Microcapsules were then produced following the same procedure described above for AC-C and PSS-C. ABTS was labeled with Hoechst 33342 to observe the interaction with the alginate microcapsules. Laser scanning confocal microscope fluorescence images of AC-C and PSS-C microcapsules with labeled enzyme were obtained using a confocal microscope FV1000 (Olympus, Tokyo, Japan). Samples were examined with a 559 nm laser as a light source for RhB excitation and Alexa fluor^®^ 405 parameters for Hoechst 33342 excitation at 40× magnification. ABTS oxidation inside the microcapsules was also observed and quantified by mixing 100 µL of AC-C or PSS-C with 100 µL of ABTS. The reaction was monitored in real time for 3 min via direct microscopic observation with a BioBlue binocular microscope (Euromex, The Netherlands), 400× magnification, 0.65 NA. The set of pictures was analyzed in light of their RGB (Red-Green-Blue color model) value inside the microcapsule to determine the percentage of the microcapsules that was occupied by oxidized ABTS. In brief, a detection algorithm was used to select the microcapsule area from each picture. The selection was transformed into a gray level image. Maximum gray level from each picture was selected and stored for further analysis. The percentage of ABTS transformation was determined by comparison with gray level after saturation.

Finally, colloidal stability was determined by measuring Zeta potential in a Nano ZS zetasizer (Malvern Instruments, Malvern, UK).

### 2.7. pH and Thermal Stability Measurements

pH stability for free and encapsulated laccase was determined by measuring activity at pH values of 2.0, 3.0, 4.0, 5.0, 6.0, and 7.0. pH solutions were prepared by adjusting the stock phosphate buffer with 1M NaOH or 1M HCl solutions. Measurements were conducted by triplicate.

Thermal stability for free and encapsulated laccase was determined by measuring activity at temperatures of 30, 40, 50, 60, 70, 80, 90, and 100 °C. Each sample was heated for 30 min prior to the measurement. Experiments were conducted in triplicate.

### 2.8. Decolorization Measurements

Decolorization of Congo Red was studied for free laccase, as well as solutions containing AC-C and PSS-C. Table 1 summarizes the main Congo Red features. A stock of artificial wastewater modeling textile industry effluents was prepared by forming a solution of 70 mg L^−1^ Congo Red in milli Q water. The stock was stored in the dark at room temperature.

Decolorization assays were conducted in 3 mL cuvettes and 100 mL solutions under dark conditions and at room temperature for 5 and 2 days, respectively. Measurements were performed by adding free or encapsulated laccase to the artificial wastewater, which contained 100 mM Congo Red. Time evolution of decolorization was followed spectrophotometrically (Genesis 10S Spectrophotometer (Thermo Fisher Scientific, Waltham, MA, USA)), by collecting spectra (350 to 700 nm) every 30 min. Assays were conducted in triplicate. The residual dye concentration for 3 mL cuvettes and pilot scale experiment of 100 mL was calculated according to Equation (1):(1)$$\%\;decolorization=\frac{A_{i}-A}{A_{i}}*100$$
where $A_{i}$ is the area under the absorbance curve between 350 and 700 nm for the untreated dye, and A is the measured area of the treated dye at each given time.

Initial reaction rates were estimated from the degradation kinetics of Congo Red. Briefly, decolorization percentages were converted to molar concentrations by considering an initial Congo Red concentration of 100 mM for enzymatic reactions conducted with both free and encapsulated enzymes. Reaction rates were subsequently determined by taking the derivative of a second-order polynomial, which was previously fitted to the time traces of Congo Red concentration. Initial reaction rates correspond to the linear kinetic regime, which was observed only for data up to 75 min.

Morphology changes in both AC-C and PSS-C were observed before and after decolorization experiments under a JSM-6490LV scanning electron microscope (SEM) (JEOL, Japan). Prior to imaging, the microcapsules were gold coated using a Desk IV sputter coater (Denton Vacuum, USA).

## 3. Results

### 3.1. Microcapsules Characterization

Microcapsules were observed under a light transmission microscope and characterized via image analysis. On average, AC-C microcapsules had a diameter of 9.23 ± 0.12 µm, while PSS-C microcapsules had a diameter of 9.25 ± 0.35 µm.

The extent of encapsulation was attained by measuring the activity of the capsules and the supernatant after the encapsulation reaction. As shown in Figure 1, the encapsulation efficiency was calculated to be 61.57% ± 1.37% $\left({U\;L}^{-1}/U_{o}{\;L}^{-1}\right)$ and 54.04% ± 6.69% $\left({U\;L}^{-1}/U_{o}{\;L}^{-1}\right)$ for the AC-C and PSS-C microcapsules, respectively. This was calculated by comparing the activities of the laccase prior to and after encapsulation. This demonstrates that enzyme encapsulation is achieved with an efficiency greater than 50% for both types of polymers. Superior efficiencies were obtained for AC-C microcapsules when compared with PSS-C microcapsules.

Free laccase assays started with 2000 U L^−1^ by diluting the stock solution containing LacI and LacII. Similarly, capsule assays were conducted at an initial activity of 2000 U L^−1^ by diluting AC-C and PSS-C in 10 mL of buffer solution at pH 4.0. PSS-C activity per mass unit (U mg^−1^) approached 70.22 U mg^−1^, while for AC-C, it was 9.38 U mg^−1^. The activity of encapsulated laccase was assured by separating and washing the capsules prior to the assay. Effective encapsulation of enzyme on AC-C and PSS-C was observed with confocal microscopy, as shown in Figure 2, where fluorescence of the labeled enzyme (Figure 2a,d) was merged with the transmittance images (Figure 2b,e) of the microcapsules.

Zeta potential was collected to determine the level of aggregation of each type of microcapsule and free laccase at pH 4.0 and 7.0. Zeta potential results are shown in Figure 3. Colloidal stability for AC-C increased by about threefold and fivefold with respect to free laccase at acid and neutral pH values, respectively. In contrast, the increase in colloidal stability of PSS-C approached twofold with respect to free laccase at both acidic and neutral pH values.

The ability of encapsulated laccase to catalyze the conversion of ABTS at pH 4.0 was determined by direct microscopy observation. As evidenced in Figure 4, after the first minute, biocatalysis of ABTS by AC-C almost doubled that of PSS-C. Diffusion of ABTS through polymeric walls of AC-C could be assured through resulting confocal microscopy studies, where labeled ABTS fluorescence was overlaid with AC-C transmission images (Figure 4c,d). Nonetheless, overall ABTS oxidation by PSS-C was 22% greater than that observed for AC-C.

### 3.2. pH and Thermal Stability

pH stability of free and encapsulated laccase was determined by measuring laccase activity in a range between pH 2.0 and pH 7.0. Free laccase showed a maximum activity at pH 3.0, and a steep decrease below 60% (oxidation) at pH values above 6.0 (Figure 5a). For encapsulated laccase, maximum activities were observed at higher pH values, specifically, in the range of 4.0 to 5.0. For pH values above 6.0 and below 3.0, the activity steeply decreased from 60% to 20% (Figure 5a).

Thermal stability of free and encapsulated laccase was determined by measuring laccase activity in a range between 30 and 100 °C. For free laccase, a maximum activity was observed at 60 °C, with an approximately linear decrease for temperatures above 80 °C and below 40 °C (Figure 5b). This trend agrees well with previously reported results [29]. For encapsulated laccase, the maximum activity shifted to temperatures of about 80 °C (Figure 5b). For temperatures above 90 °C, the activity of encapsulated laccase decreased significantly, approaching 36% and 5% for AC-C and PSS-C, respectively. At temperatures below 60 °C, PSS-C exhibited activities higher than those observed for AC-C. This difference is exacerbated at 30 °C, where PSS-C remained at approximately 65%, while ACC-C was about five times lower.

### 3.3. Decolorization Measurements

The ability of free and encapsulated laccase to discolor artificial wastewaters was tested with Congo Red solutions. Figure 6 shows the percentage of decolorization of free and encapsulated laccase as a function of time in 3 mL solutions (a), the appearance of the solutions after enzymatic treatment (b), and the absorbance of the solutions after enzymatic treatment (c). Although free and encapsulated laccase started from the same level of activity, our findings suggest that encapsulated laccase exhibited a higher decolorization rate. After five days, the percentage of decolorization leveled off. Maximum decolorization of 7.6%, 13.3%, and 27.1% was reached by free laccase, AC-C, and PSS-C, respectively.

The decolorization percentage of free and encapsulated laccase in a stock solution of 100 mL as a function of time is shown in Figure 7a. After 20 h exposure of solutions to microencapsulates, the maximum dye removal attained was 5.1%, 11.5%, and 11.7% by free laccase, AC-C, and PSS-C, respectively.

Microcapsules without encapsulated laccase were also studied to estimate the impact of the microcapsule’s material on the dye removal. After 300 min of exposure to the dyes, the removal approached about 6.5% and 6% for AC-C and PSS-C, respectively. In contrast, after the same time, AC-C and PSS-C microcapsules in the absence of enzyme showed maximum removal values of 2.5% and 4.2%, respectively (Figure 7b). This result implies that, for the case of AC-C, out of the total dye decolorization, 61.5% was most likely attained by enzyme degradation, while the remaining 38.5% can be attributed to adsorption within the porous matrix of the capsule material. For the PSS-C case, 30% is most likely owing to enzyme degradation, while the remaining 70% was adsorbed. In future work, detailed adsorption studies should be carried out with the empty encapsulates to elucidate mechanisms and to establish the interplay of experimental conditions (e.g., time, dye concentration, temperature, and ionic force) for material saturation.

The catalytic competency, as estimated from initial reaction rates, was 10.29 µM s^−1^ for free laccase, and 31.56 µM s^−1^ and 77.80 µM s^−1^ for AC-C and PSS-C, respectively.

The morphology of capsules prior to and after decolorization was evaluated via SEM imaging. Images revealed more intricate surfaces for AC-C than for PSS-C (Figure 8b,c and Figure 9b,c). They also showed that crystal structures are formed within the microcapsules most likely owing to dye degradation (Figure 8e,f and Figure 9e,f).

## 4. Discussion

Microencapsulation of laccase enhanced the pH stability of the enzyme molecules, as evidenced by the 30% increase in activity at pH values above 4. This is likely attributed to a higher adsorption of dye on the microcapsules at higher pH values, which has been reported to be beneficial for laccase catalysis [30]. Upon encapsulation, thermal stability of the enzyme molecules was improved by about 30% with respect to free laccase at temperatures above 80 °C. This is likely owing to a reduction in the heat flux from the solution to the interior of the capsule. The increased thermal resistance can be attributed to the relatively low conductivity of alginate/chitosan matrices and PSS [31,32]. Additionally, these polymers have shown high degradation temperatures in the range of 180 °C [33] to 200 °C [34]. The relatively high resistance of encapsulated laccase to alkaline environments and temperatures of around 80 °C allows their application in the biological treatment of wastewaters, where high pH values are ubiquitous and high processing temperatures might be required to enable novel routes for process intensification [30].

Encapsulated laccase increased the percentage of dye decolorization by 2 (AC-C) and 3.5 (PSS-C) times with respect to free laccase. This has been attributed to the shielding of the enzyme molecules from dye degradation byproducts, which largely inhibit laccase activity [35]. This inhibition can be considered non-competitive because the affinity of the enzyme to the dye remained unchanged throughout the assays, regardless of the steady-state decolorization values. This is evidenced by the close values of decolorization for the free and encapsulated laccases at about 11 h of reaction, which corresponds to roughly half of the steady-state values (11 h, Figure 6a). Inhibition from residual products was most likely the reason for a diminishment of the rate of catalytic activity after 20 h. In addition, the enhanced dye degradation efficiency of encapsulated laccase can be attributed to higher catalytic conversion rates, which is a direct consequence of the elevated enzyme concentration within the microcapsules. Confocal microscopy allowed imaging the enzyme distribution and monitoring the diffusion of labeled ABTS (Figure 4c) within AC-C and PSS-C microcapsules, thereby highlighting their use as effective encapsulated biocatalysts.

SEM observation prior to and after decolorization allowed direct inspection of morphological changes on the microcapsule surfaces. PSS-C microcapsules exhibited smoother surfaces compared with AC-C. This is a consequence of higher linearity of cross-linked polymer chains of PSS compared with those of alginate/chitosan matrices [36]. The decolorization process led to a shrinkage in the capsule size of both PSS-C and ACC-C, which suggests that packing of polymer chains was altered during the reaction. The presence of crystalline structures inside the microcapsules after decolorization (Figure 8e and Figure 9e) suggests that byproducts of Congo Red degradation might undergo crystallization. These structures were more abundant inside AC-C, which has been attributed to chitosan gelation processes via acidic systems, where the protonation of pendant NH_2_ along the backbone enhances crystallization [37]. This appears to provide additional evidence for a lower accumulation of residual dye byproducts inside PSS-C, which was also previously observed under the light transmission microscope experiments. Because, on a per mass basis, PSS-C contained more active enzyme molecules than AC-C, the overall increase in catalyzed ABTS for PSS-C was easily observable after 3 min.

Initial reaction rates revealed that AC-C encapsulation led to a threefold increase in catalytic competency, while PSS-C resulted in a sevenfold increase. This supports the notion that PSS-C encapsulation provides an ideal environment for catalysis, where pore size allows the removal of inhibition byproducts, thereby maintaining a relatively large population of competent enzyme molecules. Table 2 shows a comparison in terms of advantages and disadvantages of using free laccase, AC-C, and PSS-C for decolorization purposes. Overall, PSS-C appears as an attractive alternative for biocatalysis in low pH environments. Moreover, its superior catalytic competency is an advantage over free laccase samples.

Table 3 shows a comparison between our study and similar assays for encapsulated laccases from different fungi sources. Alginate/chitosan beads are generally the preferred choice for encapsulation and subsequent dye decolorization treatment. Compared with our PSS preparation, alginate/chitosan microcapsules reported in [38] degraded 2.6 times more dye, albeit with a 135-fold increase in units of enzyme per mg of dye. Similarly, poly(vinyl) alcohol microcapsules from [39] degrade 1.7 times more dye, despite an eightfold increase in units of enzyme per mg of dye. This confirms that our approach provides a higher catalytic performance than closely related encapsulates reported in the literature. Nevertheless, it must be acknowledged while making a comparison that each study found in the literature uses different amounts of enzymes from a great variety of purities, as well as different dye concentrations and reactor configurations [40].

In our experiment, similar decolorization results were obtained with 3 mL solutions and 100 mL of artificial wastewater. AC-C presented better results at higher volumes, contrary to PSS-C, which changed from 15% to 11% of decolorization after 20 h.

## 5. Conclusions

Encapsulation of fungal laccase in polymeric microcapsules showed a marked increase in the biocatalytic competency towards the degradation and removal of dye content in artificial wastewaters from the textile industry, in comparison with free laccase. Encapsulation in polystyrene sulfonate (PSS) led to maximal decolorization rate of Congo Red compared with alginate/chitosan (AC) matrices. On a per mass basis, enzymatic activity for PSC-C significantly exceeded that of AC-C. With respect to free laccase, initial reaction rates increased threefold for AC-C and sevenfold for PSS-C. The microencapsulation method reported here appears as a viable avenue for the highly-efficient biocatalytic transformation of textile effluents at low pH values and relatively high temperatures. In comparison with PSS-C, alginate/chitosan microcapsules reported in [40] decolorized 2.6 times more, although with a 135-fold increase in units of enzyme per mg of dye. Moreover, poly(vinyl) alcohol microcapsules from [37] degrade 1.7 times more dye, despite an eightfold increase in units of enzyme per mg of dye. Accordingly, compared with similar microencapsulated systems, the enzyme preparations presented here might be financially attractive for bioremediation as a significantly lower amount is needed for the same level of decolorization.

## Figures and Tables

**Figure 1 polymers-12-01353-f001:**
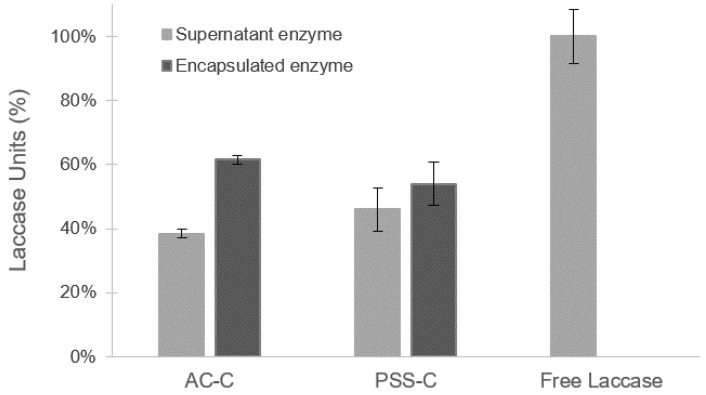
Measurement of laccase units for alginate/chitosan microcapsules (AC-C) and poly(styrenesulfonate) microcapsules (PSS-C) microcapsules and supernatant solutions in relation to initial free enzyme. Enzyme encapsulation was achieved with an efficiency greater than 50% for both types of polymers. Error bars correspond to standard deviations of experiments conducted in triplicate.

**Figure 2 polymers-12-01353-f002:**
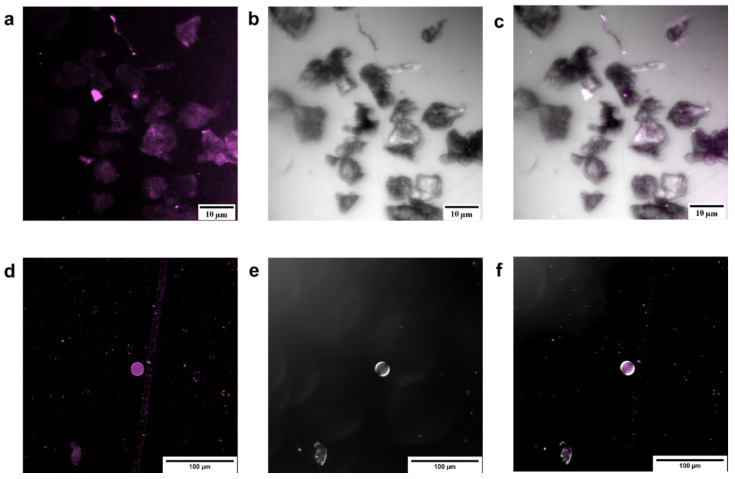
Confocal microscopy images of AC-C and PSS-C at magnification 40x. (**a**) AC-C and RhB fluorescence with 599 nm laser, (**b**) AC-C transmission image, (**c**) merging of (**a**,**b**) channels, (**d**) PSS-C and RhB fluorescence with 599 nm laser, (**e**) PSS-C transmission image, and (**f**) merging of (**d**,**e**) channels. Effective encapsulation of enzyme was achieved for AC-C and PSS-C, as demonstrated by the relatively high intensity level of fluorescence signal along the materials.

**Figure 3 polymers-12-01353-f003:**
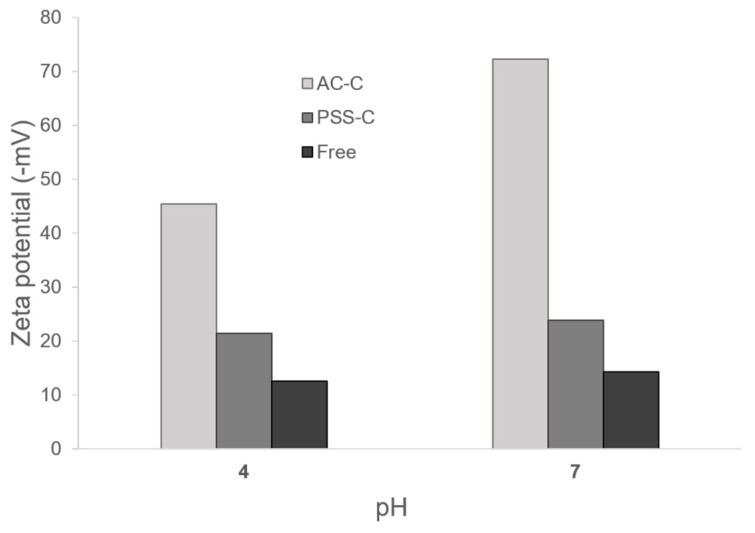
Zeta potential measurements of AC-C, PSS-C, and free laccase at pH 4.0 and 7.0. Colloidal stability for AC-C increased by about threefold and fivefold with respect to free laccase at acid and neutral pH values, respectively.

**Figure 4 polymers-12-01353-f004:**
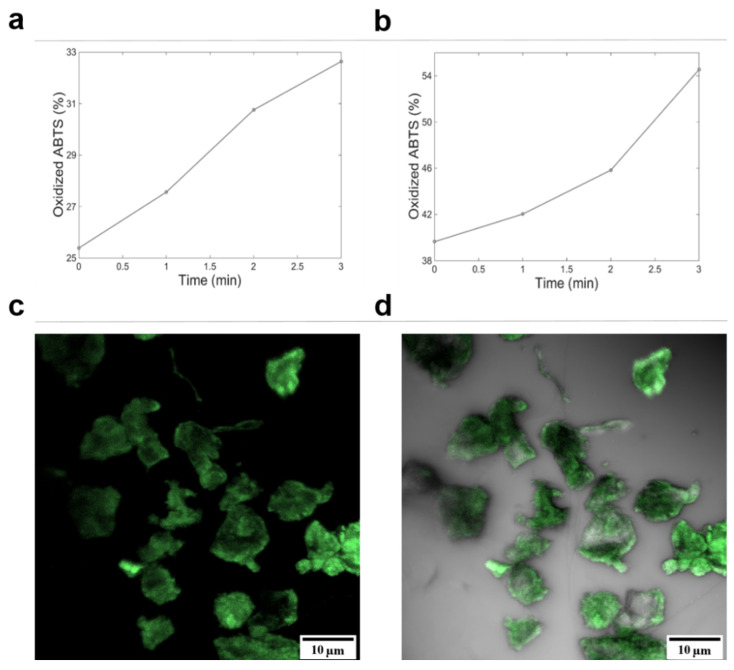
Light transmission imaging of 2,2′-azino-bis(3-ethylbenzothiazoline-6-sulfonic acid) diammonium salt (ABTS) catalyzed at pH 4.0 inside the microcapsules. (**a**) AC-C and (**b**) PSS-C, (**c**) confocal microscopy images of AC-C at magnification 40× of ABTS + Hoechst 33342 fluorescence with Alexa fluor^®^ 405 observation parameters, and (**d**) transmission image merged with (**c**) channel. Fluorescence demonstrated the ability of ABTS to penetrate the move freely across the microcapsules’ porous walls.

**Figure 5 polymers-12-01353-f005:**
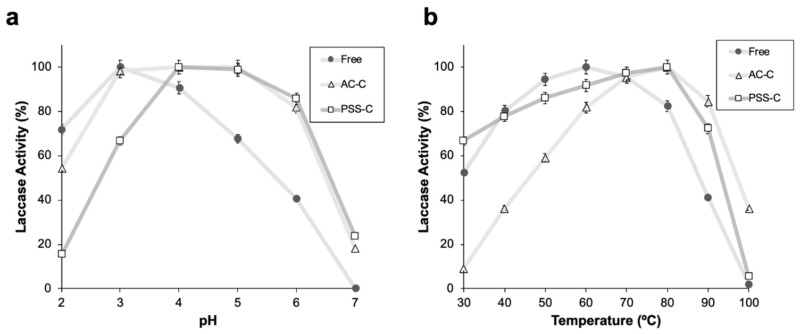
(**a**) pH and (**b**) thermal stability measurements for free laccase (●), AC-C (∆), and PSS-C (□). For free laccase, a maximum activity was observed at 60 °C, with an approximately linear decrease for temperatures above 80 °C and below 40 °C. For encapsulated laccase, the maximum activity shifted to temperatures of about 80 °C. Error bars correspond to standard deviations of experiments conducted in triplicate.

**Figure 6 polymers-12-01353-f006:**
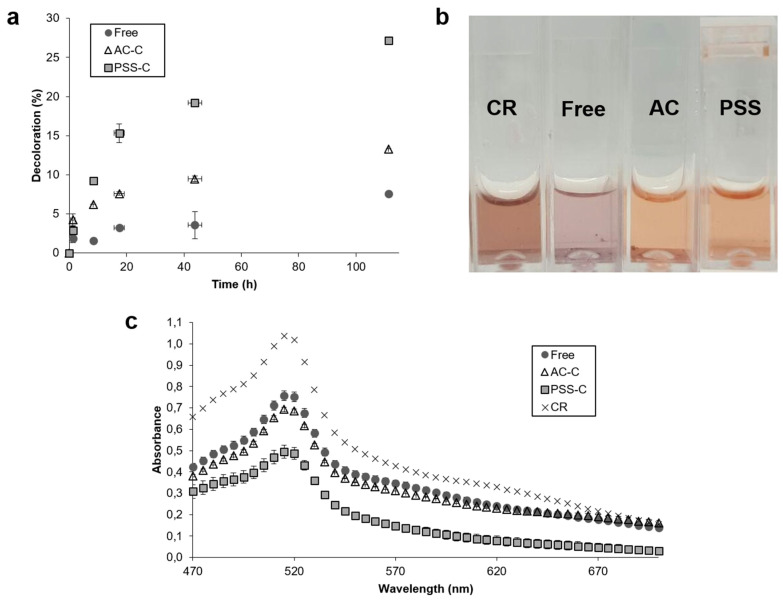
Decolorization measurements of free laccase, AC-C, and PSS-C in 3 mL cuvettes. (**a**) Percentage of decolorization as a function of time, and (**b**) solutions after decolorization assays (**c**) final spectrophotometric measurement of Congo Red (CR), free laccase, AC-C, and PSS-C. Upon encapsulation, the decolorization activity of laccase molecules significantly increased and approach about 25% for those encapsulated in PSS and about 10% for AC microcapsules. This was corroborated by colorimetric and spectrophotometric measurements.

**Figure 7 polymers-12-01353-f007:**
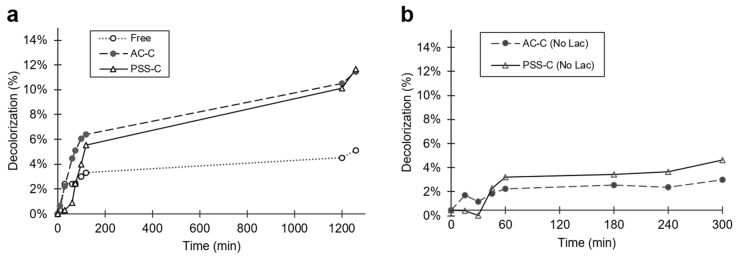
Time evolution of decolorization measurements of (**a**) free laccase, AC-C, and PSS-C and (**b**) AC-C and PSS-C without encapsulated laccase, in 100 mL Congo Red (70 mg L^−1^) batch solutions. A linear regime of superior decolorization activity during the first 2 h was followed by a regime of approximately constant activity for the subsequent 18 h.

**Figure 8 polymers-12-01353-f008:**
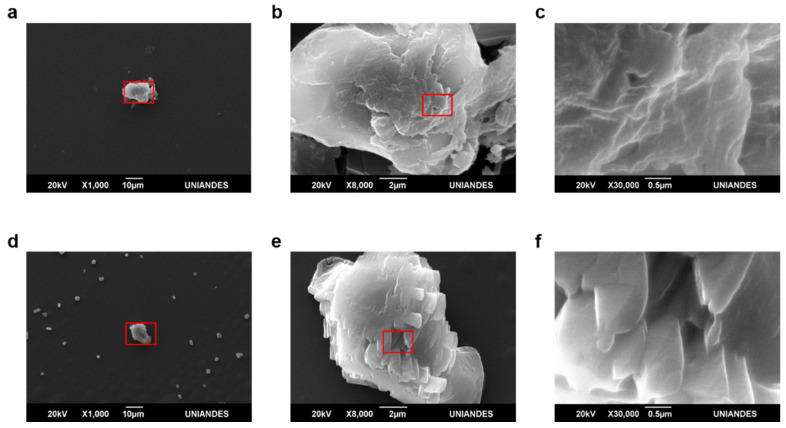
Scanning electron microscope (SEM) images of AC-C prior to and after decolorization experiments. (**a**–**c**) before decolorization of Congo Red. (**d**–**f**) after decolorization of Congo Red. Surface morphology was considerably intricate and showed terrace-like structures. Additionally, it exhibits deposits, which were explained by the formation of crystals upon dye degradation.

**Figure 9 polymers-12-01353-f009:**
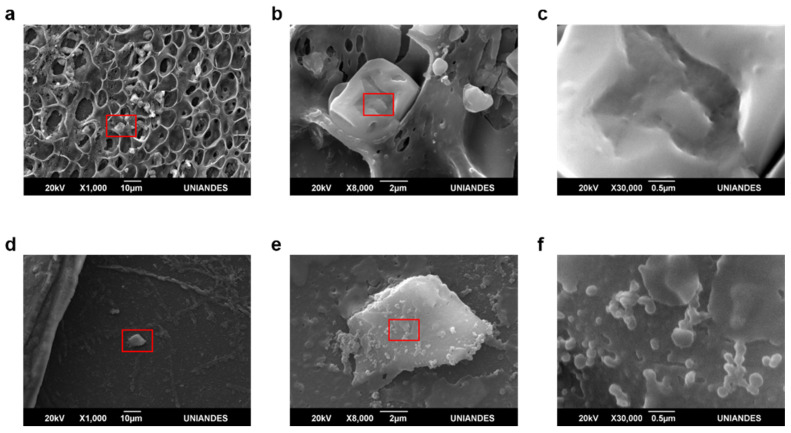
SEM images of PSS-C prior to and after decolorization experiments. (**a**–**c**) before decolorization of Congo Red. (**d**–**f**) after decolorization of Congo Red. Surface morphology was quite smooth with an absence of observable irregular features. Additionally, it exhibits deposits, which were explained by the formation of crystals upon dye degradation.

**Table 1 polymers-12-01353-t001:** Congo Red dye main features.

Dye	λ_max_ (nm)	Color Index Number	Color Index Name	Structure
**Congo Red**	500	22120	Direct Red 28	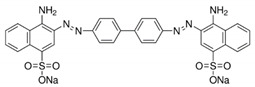

**Table 2 polymers-12-01353-t002:** Comparison between free laccase, alginate/chitosan microcapsules (AC-C), and poly(styrenesulfonate) microcapsules (PSS-C).

Treatment	Advantages	Disadvantages
**Free laccase**	Low temperature High efficiency in terms of catalytic mass Hazardous compound has the highest diffusion rate, owing to the absence of a physical barrier	High pH values Low recovery Low decolorization rate Low initial reaction rate, presumably owing to competitive inhibition Low colloidal stability Low dependence between Zeta potential and pH value
**AC-C**	High pH values High recovery Medium decolorization rate in comparison with free laccase Medium initial reaction rate (threefold increase with respect to free laccase) High colloidal stability High dependence between Zeta potential and pH value	Low temperature Medium decolorization rate in comparison with PSS-C Low efficiency in terms of catalytic mass Hazardous compound has a low diffusion rate, owing to the presence of a physical barrier
**PSS-C**	High pH values High recovery High decolorization rate Medium efficiency in terms of catalytic mass High initial reaction rate (sevenfold increase with respect to free laccase) Hazardous compound has a medium diffusion rate, owing to the presence of a physical barrier	Low temperature Low colloidal stability Low dependence between zeta potential and pH value

**Table 3 polymers-12-01353-t003:** Comparison between laccases from different fungi.

Fungi	[Dye] (mg L^−1^)	% of Decolorization	Physical State	Enzyme Activity (U L^−1^)	Enzyme Activity per Dye Concentration	Ref.
***P. Sanguineus***	70	27.1%	Encapsulated in PSS	2000	28.6 U mg^−1^	This study
***P. Sanguineus***	70	13.3%	Encapsulated in alginate/chitosan	2000	28.6 U mg^−1^	This study
***Trametes Versicolor***	10	48%	Encapsulated in poly(vinyl alcohol)	1340	134 U mg^−1^	[39]
***Pleurotus ostreatus***	31	70%	Encapsulated in alginate/chitosan	120,000	3871 U mg^−1^	[38]
***M. Thermophila***	100	3%	Encapsulated in alginate/chitosan	Not reported	-	[41]

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
