# Peer review of "Enhanced Catalytic Dye Decolorization by Microencapsulation of Laccase from *P. Sanguineus* CS43 in Natural and Synthetic Polymers"

_polymers, 2020, doi:10.3390/polym12061353_

Round 1

Reviewer 1 Report

The authors have prepared a biopolymer and polystyrene microcapsule for Laccase immobilization for decolorizing the Congo red. The authors have compared the free laccase with immobilized ones on PSS and AC-C. They have shown that the AC-C is better active somehow than PSS. However, there are some points that should be declared before publication:

-The activity of microcapsules without laccase should be studied. 

-It should be noted that what fraction is adsorption and what fraction is degradation. 

-English should be revised

-In the introduction, some recent advances should be noted in the introduction. For example, Interaction of Yarrowia lipolytica lipase in carbon-coated Fe3O4 which is reported by Prof. S. Rostamnia's group in International J. Biol. Macro. Mol.

Author Response

The original remarks from the reviewers are shown below in bold. Changes in the manuscript are explained herein and highlighted with the aid of the tracking changes tool.

Reviewer 1

The authors have prepared a biopolymer and polystyrene microcapsule for Laccase immobilization for decolorizing the Congo red. The authors have compared the free laccase with immobilized ones on PSS and AC-C. They have shown that the AC-C is better active somehow than PSS. However, there are some points that should be declared before publication:

  • The activity of microcapsules without laccase should be studied.

Changes were included in the sections 2.5. Microcapsules formation and 3.3. Decolorization measurements. Microcapsules without encapsulated laccase were fabricated and evaluated for dye decolorization under the same conditions. (see Page 3, Lines 132-133, and Page 9 and 10)

  • It should be noted that what fraction is adsorption and what fraction is degradation.

Changes were included in the section 3.3. Decolorization measurements. Results of the dye decolorization due to microcapsules with and without encapsulated laccase were compared to obtain the dye removal fraction due to adsorption within the capsules and that attributed to enzymatic degradation. (see Page 9 and 10, Line 303 - 317)

  • English should be revised

The command of the English language was revised throughout the entire manuscript

  • In the introduction, some recent advances should be noted in the introduction. For example, Interaction of Yarrowia lipolytica lipase in carbon-coated Fe3O4 which is reported by Prof. S. Rostamnia's group in International J. Biol. Macro. Mol.

Changes were included in the section 1. Introduction. Recent publications were included in the References section of the revised version of the manuscript:

  1. Anku, W.W.; Mamo, M.A.; Govender, P.P. Phenolic Compounds in Water: Sources, Reactivity, Toxicity and Treatment Methods. In Phenolic Compounds - Natural Sources, Importance and Applications; InTech, 2017.

  1. Jun, L.Y.; Yon, L.S.; Mubarak, N.M.; Bing, C.H.; Pan, S.; Danquah, M.K.; Abdullah, E.C.; Khalid, M. An overview of immobilized enzyme technologies for dye and phenolic removal from wastewater. Journal of Environmental Chemical Engineering 2019, 7, 102961.

  1. Fathi, Z.; Doustkhah, E.; Rostamnia, S.; Darvishi, F.; Ghodsi, A.; Ide, Y. Interaction of Yarrowia lipolytica lipase with dithiocarbamate modified magnetic carbon Fe3O4@C-NHCS2H core-shell nanoparticles. International Journal of Biological Macromolecules 2018, 117, 218–224.

  1. Allam, N.G.; Ismail, G.A.; El-Gemizy, W.M.; Salem, M.A. Biosynthesis of silver nanoparticles by cell-free extracts from some bacteria species for dye removal from wastewater. Biotechnology Letters 2019, 41, 379–389.

  1. Campaña, A.L.; Sotelo, D.C.; Oliva, H.A.; Aranguren, A.; Ornelas-Soto, N.; Cruz, J.C.; Osma, J.F. Fabrication and Characterization of a Low-Cost Microfluidic System for the Manufacture of Alginate–Lacasse Microcapsules. Polymers 2020, 12, 1158.

Reviewer 2 Report

In this work, the authors used polymeric microcapsules of alginate/chitosan or poly(styrenesulfonate) (PSS) to encapsulate laccase from Pycnoporus sanguineus CS43 and evaluate the impact on catalytic activity during the degradation of Congo Red. They implemented light transmission spectroscopy to monitor the diffusion of Congo Red into the microcapsules by tracking the conversion of 2,2’-azino-bis(3-ethylbenzothiazoline-6-sulfonic acid) diammonium salt (ABTS). They evaluated the thermal and pH stability of free and encapsulated laccases and Congo Red decolorization and determined Congo Red decolorization. A scanning electron microscope (SEM) was applied to observe encapsulated laccases prior and after the degradation process. They also calculated initial reaction rates for each encapsulate. This work may provide a suitable avenue to more efficient laccase dye decolorization.

In this study, because the authors used the immobilization technique to cut the cost of laccase during the degradation of Congo Red, I suggest that the reuse assay of the encapsulated laccases shall be investigated.

Author Response

The original remarks from the reviewers are shown below in bold. Changes in the manuscript are explained herein and highlighted with the aid of the tracking changes tool.

Reviewer 2

In this work, the authors used polymeric microcapsules of alginate/chitosan or poly(styrenesulfonate) (PSS) to encapsulate laccase from Pycnoporus sanguineus CS43 and evaluate the impact on catalytic activity during the degradation of Congo Red. They implemented light transmission spectroscopy to monitor the diffusion of Congo Red into the microcapsules by tracking the conversion of 2,2’-azino-bis(3-ethylbenzothiazoline-6-sulfonic acid) diammonium salt (ABTS). They evaluated the thermal and pH stability of free and encapsulated laccases and Congo Red decolorization and determined Congo Red decolorization. A scanning electron microscope (SEM) was applied to observe encapsulated laccases prior and after the degradation process. They also calculated initial reaction rates for each encapsulate. This work may provide a suitable avenue to more efficient laccase dye decolorization.

  • In this study, because the authors used the immobilization technique to cut the cost of laccase during the degradation of Congo Red, I suggest that the reuse assay of the encapsulated laccases shall be investigated.

The reuse of the microcapsules was out of the scope of this work. The method of filtration used for microcapsules separation and recovery from the media allowed us to recuperate less than 50% of the total microcapsule mas used for the dye removal experiment. The rest of mass of the capsules stayed attached to the surface of the filter and trapped within the pores of it. Assays to remove the attached and trapped microcapsules showed that recovered material presented altered morphologies and traces of fibers due to the interaction with the filter. Evaluating reusability was not feasible with this filtration methodology. Our upcoming contribution will focus on the development of more effective bioseparation methods to improve the fraction of recovered microcapsules and ultimately evaluate their reusability through residual activity measurements.

Round 2

Reviewer 1 Report

The authors have carefully addressed the comments in the manuscript. The current format is acceptable for the journal. 

Reviewer 2 Report

I have no question about this manuscript.